# Prediction of Potential Suitable Distribution Areas of *Quasipaa spinosa* in China Based on MaxEnt Optimization Model

**DOI:** 10.3390/biology12030366

**Published:** 2023-02-25

**Authors:** Jinliang Hou, Jianguo Xiang, Deliang Li, Xinhua Liu

**Affiliations:** College of Animal Science and Technology, Hunan Agricultural University, Changsha 410128, China

**Keywords:** *Quasipaa spinosa*, MaxEnt, ArcGIS, potential habitat, environment variable

## Abstract

**Simple Summary:**

In this study, in order to understand the distribution and optimal living environment of *Quasipaa spinosa* in China, we made predictions about the impacts of different environmental climates on its habitat. Our results show that our model is highly reliable. The mountainous areas in southern China with sufficient water supply are the main suitable areas for this species. The future reduction in emission concentration will be friendly to the ecological environment in the suitable areas of this species, better protecting the reproduction of its natural population.

**Abstract:**

*Quasipaa spinosa* is a large cold-water frog unique to China, with great ecological and economic value. In recent years, due to the impact of human activities on the climate, its habitat has been destroyed, resulting in a sharp decline in natural population resources. Based on the existing distribution records of *Q. spinosa*, this study uses the optimized MaxEnt model and ArcGis 10.2 software to screen out 10 factors such as climate and altitude to predict its future potential distribution area because of climate change. The results show that when the parameters are FC = LQHP and RM = 3, the MaxEnt model is optimal and AUC values are greater than 0.95. The precipitation of the driest month (bio14), temperature seasonality (bio4), elevation (ele), isothermality (bio3), and the minimum temperature of coldest month (bio6) were the main environmental factors affecting the potential range of the *Q. spinosa*. At present, high-suitability areas are mainly in the Hunan, Fujian, Jiangxi, Chongqing, Guizhou, Anhui, and Sichuan provinces of China. In the future, the potential distribution area of *Q. spinosa* may gradually extend to the northwest and north. The low-concentration emissions scenario in the future can increase the area of suitable habitat for *Q. spinosa* and slow down the reduction in the amount of high-suitability areas to a certain extent. In conclusion, the habitat of *Q. spinosa* is mainly distributed in southern China. Because of global climate change, the high-altitude mountainous areas in southern China with abundant water resources may be the main potential habitat area of *Q. spinosa*. Predicting the changes in the distribution patterns of *Q. spinosa* can better help us understand the biogeography of *Q. spinosa* and develop conservation strategies to minimize the impacts of climate change.

## 1. Introduction

The geographical distribution of species is the result of adaptation to climate, topography, soil, biology, migration history, and human activities during the long evolutionary process, which reflects the evolutionary history of species, population expansion, and the ability to adapt to the environment [1]. In recent years, habitat destruction due to human disturbance and global climate change has led to the migration, rapid decline, or disappearance of amphibian species [2]. Therefore, it is of great significance to study the suitable areas of species to protect their habitats in situ. The application of niche models to predict and evaluate species habitats is becoming a new hot topic [3,4]. There are more than 10 species distribution models (SDMs) that have been reported, but the MaxEnt model is low cost, simple to operate, short to run, and can simulate the fitness range of species well with a very small number of samples (*n* ≥ 5) [5]. At present, it is also widely used in the prediction research of amphibian habitats, such as *Odrrana hainanensis* [2], and five species of *Scutiger* [6], and others such as *Rana Zhenhaiensis* [7], *Buergeria oxycephala* [8], *Nanorana parkeri* [9], *Plethodon* [10], and *Quasipaa boulengeri* [11], and so on.

*Quasipaa spinosa*, commonly known as stone frog, rock frog, etc., belongs to *Anura*, *Dicroglossidea*, and *Quasipaa* [12]. As a large frog, it breeds in cold water streams and is mainly distributed in the hills of southern China and mountainous areas of northern Vietnam [12,13,14,15,16], which has a special ecological and biological status. Due to overfishing, the wide application of chemical pesticides and environmental changes, the wild population of *Q. spinosa* has decreased dramatically [17]. It has been designated as a vulnerable species by the International Union of the Conservation of Nature (IUCN) Red List and the Red List of Chinese Species [18].

As an important economic frog with high nutritional value and homology of medicine and food, *Q. spinosa* is honored as the “King of Hundred Frogs” [15,19,20]. In the last ten years, the market demand of *Q. spinosa* has been rising, and the price of *Q. spinosa* has increased 20–30 times from 1980s to now [18], which has promoted the rapid development of *Q. spinosa* breeding in China, becoming one of the most important industries for poverty alleviation in mountainous areas. Although artificial breeding has relieved the pressure of hunting for the wild population to some extent, the effect has been limited [21]. At present, research on *Q. spinosa* focuses on gut microbiology [13,14,22], diseases [23,24], genetic transcriptomes [25,26], and captive breeding [16,27,28,29]. It is very important to better protect and reasonably utilize the existing wild *Q. spinosa* resources to achieve a win–win situation for environmental protection and resource utilization. These have now become a new hot spot in fisheries resources research.

In this paper, we applied a MaxEnt optimization model and ArcGIS technology to comprehensively analyze the key environmental factors affecting the distribution of *Q. spinosa*, with the aim of understanding the current distribution of its habitat suitability, exploring its habitat range, and predicting its potential geographical distribution in order to provide reference for the conservation of *Q. spinosa* diversity, habitat restoration, and biogeography research.

## 2. Materials and Methods

### 2.1. Data Collection and Processing

The distribution points information of *Q. spinosa* in this study comes from the Global Biodiversity Information Facility (GBIF; https://www.gbif.org/, accessed on 6 March 2022) [30]. After retrieval, a total of 1779 distribution records of *Q. spinosa* was obtained. Through https://jingweidu.51240.com/, the longitude and latitude information of *Q. spinosa* was determined, and the data without detailed geographic location, duplicate specimens, and location deviations were excluded. Finally, 130 geographical distribution points of *Q spinosa* were screened. We converted the obtained sample point data into .csv format and used ArcGIS 10.2 to draw a map of *Q. spinosa* distribution in China (Figure 1).

Human footprints data were downloaded from the Socioeconomic Data and Applications Center (SEDAC, https://sedac.ciesin.columbia.edu, accessed on 22 April 2022) [31]. Altitude and 19 bioclimatic variables in current (1970–2000), future 2050s (2041–2060), and future 2070s (2061–2080) were downloaded from WorldClim (http://www.worldclim.org/). The future (2050s, 2070s) climatic variables were selected from two of the four greenhouse gases (RCP2.6 and RCP8.5) under the general climate system mode proposed in the fifth assessment report of IPCC—AR5—representing the lowest and highest scenarios of greenhouse gas emissions [32]. The Chinese province boundary map was downloaded from the Institute of Geographic Sciences and Natural Resources Research, Chinese Academy of Sciences (http://www.resdc.cn/). All data have a unified resolution of 2.5 min, the coordinate system was WGS 1984, and was converted to “asc” with ArcGIS 10.2 software (Environmental Systems Research Institute, Inc., America).

In order to minimize the bias fitting of the model, the ENMTools package (Package) was used to perform correlation analysis on 21 environmental factors [33]. When the correlation was greater than |0.8|, the variables with small contribution rates were removed. Finally, 10 significant factors were screened out (Table 1).

### 2.2. Species Distribution Modeling, Optimization, and Evaluation

Optimized maxent model was implemented in the R programming environment using the package ENMeval [34]. We set 5 regularization multiplier values (β, 1, 2, 3, 4, and 5) and adopted 6 features (H, L, LQ, LQH, LQHP, and LQHTP), among which L, Q, H, P, and T were linear, quadratic, hinge, product, and threshold, respectively [35,36]. Finally, the best model parameter combination with the lowest Akaike Information Criterion Correction (AICc) value (delta. AICc = 0) was obtained for the establishment of MaxEnt model.

*Q. spinosa* distribution occurrence data and selected environmental variables were loaded into the MaxEnt model. The data were randomly divided: 75% of the location point data were used as a training model, with the remaining 25% used for validating the MaxEnt model [37,38]. By default, the maximum number of iterations was set as 10,000, the model was repeated for 10 times, and the predicted results were output in “Cloglog” format and “asc” file type [39]. The maximum number of iterations after optimization is 420. The prediction effect was tested by AUC under the receiver operating characteristic curve (ROC) [40]. AUC value between 0.5 and 0.6 was considered to be unqualified, 0.6~0.7 was poor, 0.7~0.8 was average, 0.8~0.9 was good, and 0.9~1.0 was excellent [34]. When the TSS value was <0.4, the predictive power of the model was considered “poor”, while 0.4–0.8 was considered “good” and 0.8–1 was considered “excellent” [41].

### 2.3. Classification of Suitable Living Grade of Q. spinosa

The average value of MaxEnt simulation results in each period was imported into ArcGIS 10.2 software, and the model distribution area was divided into suitability grades and visualized [42]. The habitat suitability index was divided into four levels: high-suitability area, medium-suitability area, low-suitability area, and non-suitable area by the Nature Breaks (Jenks) method. They were represented by different colors: red is high-suitability area, yellow is medium-suitability area, green is low-suitability area, and white is non-suitable area. We used ArcGIS raster tools to count the area of suitable areas.

## 3. Results

### 3.1. Model Optimization and Accuracy Evaluation

The MaxEnt model was used to simulate the potential distribution areas in different scenarios currently and in future (2050s and 2070s) according to the AICc (the minimum information criterion) for the screened 130 *Q. spinosa* distribution points and 10 environmental variable layers. The optimal parameter combination of FC = LQHP and RM = 3 is obtained after the optimization of the ENMeval program package. At this time, AICc is the smallest and delta.AICc = 0. The optimized AICc and the difference between the AUC value of the training set and the AUC of the test set are both lower than the default settings (Table 2).

The MaxEnt simulations for modern and future (2050s and 2070s) time periods with optimal parameters are performed, and the AUC results are shown in Figure 2. The values of the area under the receiver operating characteristic curve were all greater than 0.95 (Table 3). The true skill statistics were all greater than 0.8 (Table 3). In conclusion, the optimized MaxEnt model has high reliability in the prediction results of the potential suitable areas of *Q. spinosa*. The 10 environmental factors used for modeling included 8 climate factors, 1 terrain factor, and 1 anthropogenic factor (Table 1).

### 3.2. The Importance of Environmental Variables

Percent contribution (PC) and permutation importance (PI) are the main indicators for evaluating the importance of environmental variables. The larger the index value, the higher the importance of environmental variables. The top 5 environmental variables with percent contributions are precipitation of the driest month (bio14, 79.3%), elevation (ele, 5.5%), mean diurnal range (bio2, 3.9%), isothermality (bio3, 3.4%), and temperature seasonality (bio4, 2.6%), accounting for 94.7% in total. The top 5 environmental variables with permutation importance are temperature seasonality (bio4, 40.9%), isothermality (bio3, 20.0%), precipitation of the driest month (bio14, 11.7%), minimum temperature of the coldest month (bio6, 9.6%), and elevation (ele, 8.2%), accounting for 90.4% of the total.

The Jackknife test results show (Figure 3F–H) that test gain, regularized training gain, and AUC are basically the same without variables and with all variables. When only variables are used, the top five factors with the largest test gain, regularized training gain, and AUC are precipitation of the driest month (bio14), mean diurnal range (bio2), minimum temperature of the coldest month (bio6), precipitation of the warmest quarter (bio18), and temperature seasonality (bio4), indicating that these five environmental factors have the greatest impact on the distribution of *Q. spinosa*. In conclusion, the precipitation of the driest month, temperature seasonality, elevation, isothermality, and minimum temperature of the coldest month are the dominant variables affecting the distribution of *Q. spinosa*.

The environmental variables (>0.63) were the most suitable for the survival of *Q. spinosa* when the precipitation of the driest month was 28.4–166.1 mm, the altitude was 79.6–635.8 m, the minimum temperature of the coldest quarter was 0.1–9.0 °C, the temperature seasonality was 584.0–821.3, and the isothermality was 21.2–28.0 (Figure 3A–E).

### 3.3. Current and Future Potential Suitable Areas and Their Spatiotemporal Changes

The distribution range of the potential suitable habitats of *Q. spinosa* in the future is generally the same, mainly in Hunan, Hubei, Guangxi, Fujian, Zhejiang, Anhui, Jiangxi, Guizhou, Guangdong, Sichuan, Taiwan, Hong Kong, and other provinces (Figure 4a–e). According to the MaxEnt simulation results, the modern potential total suitable area of *Q. spinosa* is 1.65 × 10^6^ km^2^, accounting for 17.19% of the total land area in China, including the high-suitability area of 4.9 × 10^5^ km^2^, accounting for 5.10%; the medium-suitability area of 6.3 × 10^5^ km^2^, accounting for 6.56%; and the low-suitability area of 5.3 × 10^5^ km^2^, accounting for 5.52% (Table 4; Figure 4f). In the future 2050s, the proportion of the total suitable area for the frog under high-concentration (RCP8.5) and low-concentration (RCP2.6) emission scenarios will be 17.92% and 18.33%, respectively. In the future 2070s, the proportions of the total suitable area of *Q. spinosa* under the high-concentration (RCP8.5) and low-concentration (RCP2.6) emission scenarios will be 17.29% and 17.71%, respectively, which are higher than the modern 17.19% (Table 4). In the future 2050s and 2070s, the proportion of suitable areas under the high-concentration emission scenario (RCP2.6) is larger than that under the low-concentration emission scenario (RCP 8.5).

Compared with the current time period, the proportion of high-suitability areas in the 2050s and 2070s was reduced. Under the future low-concentration emission scenario (RCP2.6), the area of the low-suitability area first increased and then decreased, but was higher than that of the present time; the area of the medium-suitability area was the smallest in 2050s and the largest in 2070s; and the high-suitability area showed a decreasing trend. Under the future high-concentration emission scenario (RCP8.5), the areas of medium- and low-suitability areas both increased first and then decreased but were higher than that of the current generation; and the areas of high-suitability areas decreased first and then increased, but were lower than that of the current generation.

Compared with the current potential distribution range of *Q. spinosa*, the newly added potentially suitable areas of 2050s-RCP2.6, 2050s-RCP8.5, 2070s-RCP2.6, and 2070s-RCP8.5 were all larger than the lost areas, increasing by 60,000, 20,000, 130,000, and 80,000 km^2^, respectively, accounting for 3.64%, 1.21%, 7.88%, and 4.85% of the modern area, respectively (Table 5). In the future, the potential suitable area of *Q. spinosa* showed a trend for migration to the northwest and north, and the potential suitable area in the south gradually decreased. In the future 2050s and 2070s, the loss areas are mainly in Yunnan, Guizhou, Guangdong, Taiwan, Jiangsu, and other provinces, and the newly added areas are mainly in Sichuan, Gansu, and Qinling (Figure 5). Under the future low-concentration emission scenario (RCP 2.6), Yunnan is both the main increase area and the main loss area of suitable habitat. Under the high-concentration emission scenario (RCP 8.5), more suitable habitats of *Q. spinosa* were lost in Yunnan, and the loss area increased with time (Figure 5).

Under the low-concentration emission scenario (RCP2.6), the increase in the area of suitable areas increased from 3.64% in the 2050s to 7.88%. Under the high-concentration emission scenario (RCP8.5), the area ratio increases from 1.21% in the 2050s to 4.45% in the 2070s. Under the low-concentration emission scenario in the future 2070s, the area of suitable habitats will increase the most (Table 5).

## 4. Discussion

### 4.1. Rationality of Model

Currently, among the reported species distribution models, the MaxEnt model has better stability and higher accuracy, and has less distortion in dealing with group temperature and precipitation factors [43,44,45]. Amphibians are ectothermic animals, and the external environment, especially temperature, precipitation, and so on, is the main restricting factor for their growth and development [46,47,48]. In this paper, temperature, altitude, and precipitation related to the characteristics of amphibians are selected as the main climatic factors, and climatic extreme values are selected to make related predictions. This study uses the ENMeval program package to optimize MaxEnt to reduce the degree of overfitting and sampling bias, improving the prediction accuracy [49,50]. The model prediction results shows that the AUC and TSS values of the current and future time periods (2050s and 2070s) are all greater than 0.95 and 0.80, respectively, indicating the high accuracy and distinguishing ability of the prediction results. The results of the optimized MaxEnt model showed that the potential suitable habitats of the vulnerable species *Q. spinosa* were mainly in the provinces south of the Qinling Mountains in China, which was basically consistent with the reported distribution range of *Q. spinosa* [51,52,53,54].

### 4.2. Main Environmental Factors Affecting the Distribution of Q. spinosa

*Q. spinosa* is a water-dwelling frog [55], which mainly inhabits the areas beside streams in mountainous areas, and temperature and water sources have an important impact on its survival [56]. Altitude is closely related to temperature changes to a certain extent. The results of the Jackknife, AUC, PC, and PI tests indicated that the importance of precipitation in the driest month, temperature seasonality, elevation, isothermality, and the mean temperature of the coldest quarter played a major role in affecting the distribution of *Q. spinosa*. This is similar to *Rana hanluica* [57] and *Buergeria oxycephala* [8], but the precipitation factor has a lower gain on the distribution model of *Buergeria oxycephala*, which may be related to the lower water dependence of its adult frog habitat. The distribution of *Odrrana hainanensis* is less affected by temperature, humidity, and sunshine, which may be related to the fact that its study area is located in the equatorial tropics [2]. While *Q. spinosa* is a cold-water frog with a small suitable temperature range and a distribution area mainly located in the subtropical region, large changes in temperature and precipitation have a great impact on its distribution. The main factors influencing their potential distribution varied among the five species of *Scutiger* [6]. The precipitation of the warmest quarter and temperature seasonality are the main factors affecting the potential distribution of *Rana heaviness* [7]. In summary, different species have different ecological requirements, and the main environmental factors affecting their potential distribution are also different.

This study is basically consistent with the findings of Zou et al. [58]. The results show that the most suitable habitat altitude of *Q. spinosa* is 79.6–635.8 m. However, Liu [55] found that the distribution range of *Q. spinosa* was 150–1000 m above sea level, while Fei et al. [59] found that it was 300–1500 m, and Liang et al. [60] found that it was 700–800 m. The reason may be related to climate change in recent years, sample bias, and different sampling intensities. When the altitude was lower than 79.6 m or higher than 635.8 m, the suitability of *Q. spinosa* decreased dramatically. The lower elevations in the south are mainly plains, where human dwellings congregate, and water is plentiful, but summer temperatures are high. Higher altitudes, however, are not conducive to the survival of *Q. spinosa* due to the low temperatures and low biodiversity combined with the lack of water and food. From the 1950s to 2007, the annual precipitation in the central region of South China decreased year by year, and the temperature showed an upward trend [61]. The changes in temperature and precipitation were related to the altitude [62,63,64], which led to the change in the habitat altitude of *Q. spinosa*. When the mean diurnal range is 3.5–7.7 °C and the max temperature of the warmest month is 29.5–34.3 °C (Figure 6), it is most suitable for the survival of *Q. spinosa*, which is basically consistent with the reports [16,51]. Southern China is dominated by a subtropical monsoon climate, with rain and heat in the same period. The *Q. spinosa* is optimally distributed in areas with a warmest quarter precipitation of 549.5–850.9 mm, which is basically consistent with the precipitation climate in southern China [61].

In this study, the human footprint refers to the pressure humans put on the environment. This factor has little effect on the distribution of *Q. spinosa*, which may be due to the fact that the human capture factor is not included in this environmental data and the living area of *Q. spinosa* is mainly near mountain streams.

### 4.3. Changes in Potential Suitable Areas

Compared with the current time period, under the two emission scenarios in the future, the area of low-suitability areas increased more than that of high-suitability areas, and the area of potentially suitable areas showed an overall upward trend. Under the low-concentration emission scenario, the area of suitable habitat for *Q. spinosa* showed an increasing trend, and maintaining the low-concentration emission scenario would expand the suitable habitat distribution range of *Q. spinosa*; however, the high-suitability area gradually decreased. Under the high-concentration emission scenario, the size of the suitable area first increased and then decreased. Although the area of the high-suitability area increased in the 2070s, it was still smaller than the current time period.

The prediction results of the optimization model showed that the distribution area of *Q. spinosa* tended to expand to the north and northwest in the future, which is mainly reflected in the increase in the potentially suitable areas in Qinling, Sichuan, Gansu, and other provinces. During the past century of climate change in China, the eastern region showed a decreasing trend for precipitation as a whole [65], the accumulated temperature in the southern region showed an increasing trend, the high-accumulated-temperature region gradually decreased from the southeast to the northwest, and the annual precipitation in the central region of the south decreased year by year [61]. *Q. spinosa* is a cold-water frog with an optimal survival temperature of 22–26 °C, and 80% of its life cycle is in water. Mountainous areas have good water retention, with the reduction of ice and snow providing water replenishment, which explains the expansion direction of the future suitable area for *Q. spinosa* and the reduction in high-suitability areas.

### 4.4. Resource Conservation of Q. spinosa

Based on the goal of striving to achieve peak carbon by 2030 and carbon neutrality by 2060 (referred to as the “dual carbon” goal) [66], China’s energy structure is continuing to transition to green energy. Although carbon emissions continued to grow for the fourth consecutive year, with an increase of 0.6%, the carbon emission intensity decreased by 1%, and the overall carbon emission per unit of GDP showed a downward trend and is now only 0.7 (2018) [67]. For carbon emissions trading, relevant policies have been issued to standardize the operation of the carbon market [68]. From the perspective of economic and social development status and energy structure, China’s carbon neutrality goal still faces many challenges and uncertainties [69]; however, China is committed to improving the ecological environment and promoting harmonious coexistence between man and nature. China’s carbon peak by 2030 and carbon neutrality by 2060 can be expected.

Global warming, the increase in harsh climates, and man-altered mountains and rivers have greatly affected the ecological balance of biological habitats. *Q spinosa* feeds on live insects [27,70], and climate change may indirectly affect the survival and distribution of its prey organisms, leading to the frog migrating to more suitable habitats. Comprehensive analysis shows that the changes in environmental climate since modern times have caused some changes in the habitat of *Q. spinosa*, and it tends to be in areas with an altitude of about 600 m, small temperature differences, and excellent water sources.

The following suggestions should be helpful for the conservation of *Q. spinosa* resources. First of all, under the “dual carbon” goal, the state should further strengthen the coupled management of resources and establish and improve a sustainable forest resource management mechanism, which is of great significance to further improve the natural ecosystem. Secondly, the state should put *Q. spinosa* on the list of protected animals in China, prohibiting the sale and capture of wild *Q. spinosa*. Thirdly, the state should further clarify the prohibited types and prohibited areas of pesticides, or limit the amount of use. Finally, we need to take multiple measures to protect the natural population of *Q. spinosa*, such as in situ protection, ex situ protection, proliferation and release, etc. Priority protection can only be achieved when the whole of society works together.

## 5. Conclusions

In this study, the MaxEnt model for *Q. spinosa* provided satisfactory results. The prediction results can provide an important reference for wild *Q. spinosa* development and protection. Frog diversity conservation and biogeographical research are enriched by our results. *Q. spinosa* is mainly distributed around the streams in the mountains of southern China. Climate is one of the important factors for the distribution of *Q. spinosa.* In the future, maintaining a low-carbon life could alleviate climate deterioration and protect the habitat of spiny chestnut frogs. In addition, improving mountain and wildlife protection strategies is also important.

## Figures and Tables

**Figure 1 biology-12-00366-f001:**
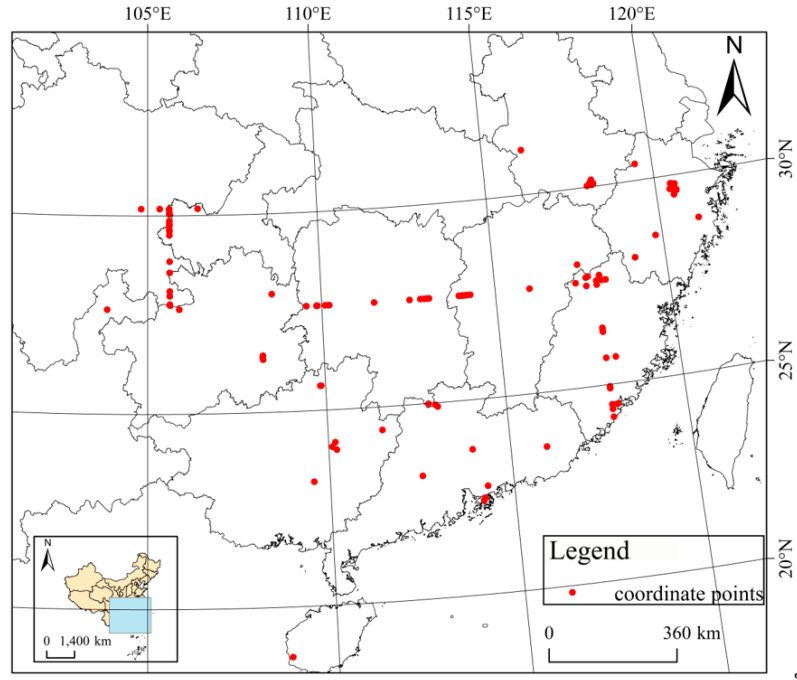
Confirmed coordinate distribution of *Quasipaa spinosa* in China.

**Figure 2 biology-12-00366-f002:**
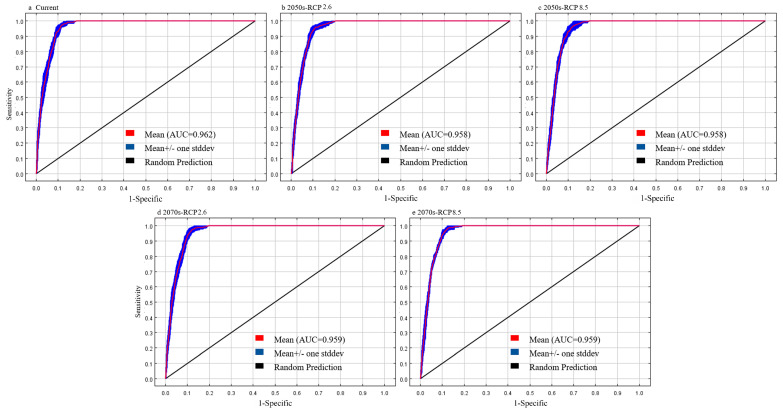
AUC value of *Q. spinosa* predicted by MaxEnt model.

**Figure 3 biology-12-00366-f003:**
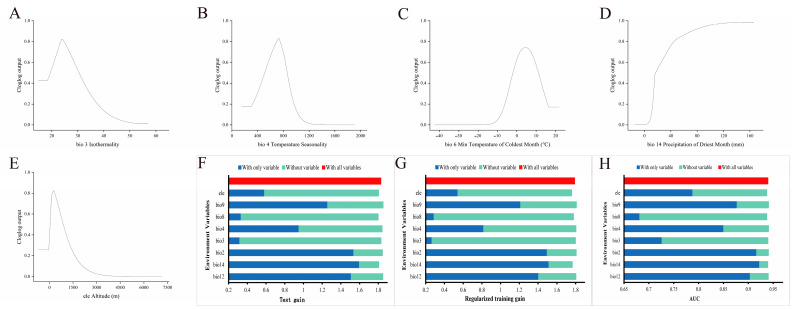
Response curves and Jackknife test of the environment variables. (**A**–**E**) Respectively represent the response curves of isothermality, temperature seasonality, min temperature of coldest month, precipitation of driest month and Altitude. (**F**–**H**) The contribution of each environmental factor to each scenario using the Jacknife test in In test gain, regularized training gain and AUC, respectively.

**Figure 4 biology-12-00366-f004:**
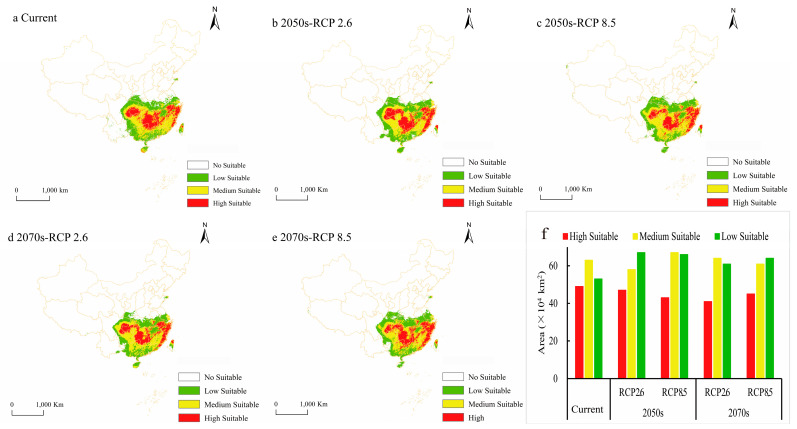
Potential distribution area of *Q. spinosa* under different scenarios. (**a**–**e**) Predicted distribution map of different suitable areas of *Q. spinosa* at current, 2050s and 2070s. (**f**) Statistical maps of different suitable areas of *Q. spinosa* in China at different periods.

**Figure 5 biology-12-00366-f005:**
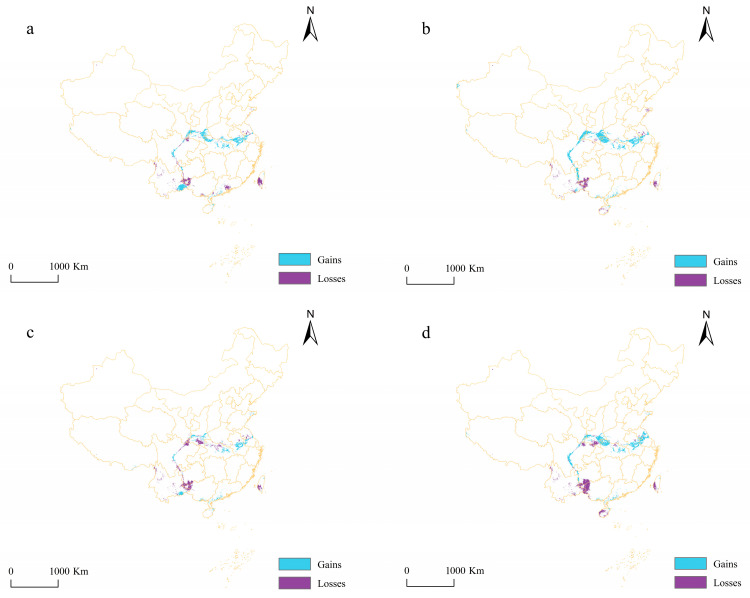
Spatial transformation pattern of *Q. spinosa* suitable area in different periods. (**a**) Greenhouse gas emission concentration is lowest in 2041–2060; (**b**) greenhouse gas emission concentration is highest in 2041–2060; (**c**) greenhouse gas emission concentration is lowest in 2061–2080; (**d**) greenhouse gas emission concentration is highest in 2061–2080. The increase and loss of suitable areas are derived and compared to current area.

**Figure 6 biology-12-00366-f006:**
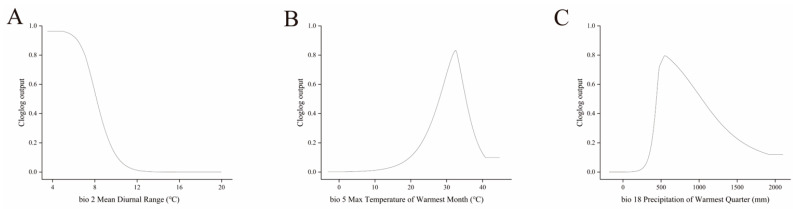
Response curve of the partial environment variables. (**A**–**C**) Respectively represent the response curves of mean diurnal range, max temperature of warmest month and precipitation of warmest quarter.

**Table 1 biology-12-00366-t001:** Various parameters of the main environmental variables of *Q. spinosa*.

Environmental Variables	Description	Unit	PC (%)	PI (%)
ele	Elevation	m	5.5	8.2
bio2	Mean Diurnal Range	°C	3.9	3.2
bio3	Isothermality	-	3.4	20.0
bio4	Temperature Seasonality	-	2.6	40.9
bio5	Max Temperature of Warmest Month	°C	0.6	0
bio6	Min Temperature of Coldest Month	°C	1.9	9.6
bio14	Precipitation of Driest Month	mm	79.3	11.7
bio15	Precipitation Seasonality	-	1.6	2.9
bio18	Precipitation of Warmest Quarter	mm	0.7	1.9
people	Human Foot	-	0.5	1.7

**Table 2 biology-12-00366-t002:** Evaluation metrics of MaxEnt model generated by ENMeval.

Type	FC	β	delta. AICc	avg. diff. AUC
Default	LQHPT	1	101.9003	0.1211
Optimized	LQHP	3	0	0.1008

FC: feature combination; RM: regulatory multiplier; AICc: LQPH: linear features (L) + quadratic features (Q) + product features (P) + hinge features (H); LQH: linear features (L) + quadratic features (Q) + hinge features (H); delta. AICc: the minimum information criterion AICc value; avg. diff. AUC: difference between the AUC value.

**Table 3 biology-12-00366-t003:** Evaluate the AUC and TSS index values of the MaxENT model.

	Current	50s-RCP2.6	50s-RCP8.5	70s-RCP2.6	70s-RCP8.5
AUC	0.962 ± 0.0073	0.958 ± 0.0073	0.958 ± 0.0073	0.959 ± 0.0072	0.959 ± 0.008
TSS	0.817 ± 0.014	0.813 ± 0.009	0.804 ± 0.010	0.822 ± 0.015	0.818 ± 0.015

Note: The values in the table are expressed as “mean ± standard deviation”.

**Table 4 biology-12-00366-t004:** Area statistics of suitable areas for *Q. spinosa* under different climate scenarios.

Circumstances	Low Suitability		Medium Suitability		High Suitability		All	
	Area (×10^4^ km^2^)	Percentage (%)	Area (×10^4^ km^2^)	Percentage (%)	Area (×10^4^ km^2^)	Percentage/%	Area (×10^4^ km^2^)	Percentage (%)
Current	53	5.52	63	6.56	49	5.10	165	17.19
2050s-RCP 2.6	67	6.98	58	6.04	47	4.90	172	17.92
2050s-RCP 8.5	66	6.88	67	6.98	43	4.48	176	18.33
2070s-RCP 2.6	61	6.35	64	6.67	41	4.27	166	17.29
2070s-RCP 8.5	64	6.67	61	6.35	45	4.69	170	17.71

Note: The area percentages are the ratios of the suitable areas of different grades to the national land area (960 × 10^4^ km^2^) under different climate scenarios.

**Table 5 biology-12-00366-t005:** Spatial variations in suitable habitat for *Q. spinosa* in different periods.

Circumstances	Area (×10^4^ km^2^)			Rate of Change (%)		
	Gain	Loss	Change	Gain	Loss	Change
2050s-RCP2.6	14	8	6	8.48	4.85	3.64
2050s-RCP8.5	9	7	2	5.45	4.24	1.21
2070s-RCP2.6	18	5	13	10.91	3.03	7.88
2070s-RCP8.5	14	6	8	8.48	3.64	4.85

Note: The rate of change is the percentage of the area of each period and the area of the contemporary suitable area. The current *Q. spinosa* area of the potential suitable habitat is 165 × 10^4^ km^2^.

## Data Availability

All data generated by this study are available from the corresponding author upon reasonable request.

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
