# Peer review of "Prediction of Potential Suitable Distribution Areas of Quasipaa spinosa in China Based on MaxEnt Optimization Model"

_biology, 2023, doi:10.3390/biology12030366_

Round 1

Reviewer 1 Report

The manuscript entitled "Prediction of potential suitable distribution areas of Quasipaa spinosa in China based on MaxEnt optimization model” used a distribution modeling approach together with relevant environmental variables to predict current and future distribution pattern of Quasipaa spinosa in China. Although similar methodologies are common, results of the study could have useful implications for management actions. However, the work requires major changes before its ready for publication:

Comments and suggestions:

Introduction:

- Introduction does not provide sufficient review of the available geospatial modeling techniques, for example, available SDMs, particularly for the target species in the context of this study. In addition, previous studies on the target species on the context of this study has not been reviewed.

- Objectives of the study should be clearly spelled out please.

2. Materials and Methods

2.1. Data collection and processing

- The full link of the GBIF should be provided from where the records were

 downloaded.

- It’s not clear whether any spatial filtering or 'thinning' was performed for the presence records? In other word, what was the minimum distance between to occurrence points?

- Why Phase 5 (CMIP 5) was used in this study?  According to the “worldclim” website, the Representative Concentration Pathways (RCPs) are now obsolete and they recommend the use of the new version (CMIP6 data; i.e., the Shared Socio-economic Pathways (SSPs) in similar studies.

2.2. Species distribution modeling, optimization and evaluation

-  "5 regularization multiplier value" this should be expressed as “(β)”

- “by default, the maximum number of iterations is set as 10000, “since the model was optimized, was more than or less than 10000 background points were tested?

- "AUC" Sometimes AUC alone is not sufficient to evaluate the model performance. Why TSS was not considered alongside the AUC?

3. Results

- Please refine/recreate the Figures 2, 3, and 5, hard to read.

- All the Figures need refinement please.

- I also suggest moving Figure 2 to the appendix or supplementary document.

4. Discussion

- The discussion section here in this paragraph is superficial and requires more work. Particularly when it comes to the in depth explanation of the key factors influencing the distribution of the target species. In other words, how environmental variables influence the spatial distribution of the target species is required. In addition, the discussion section requires drawing parallels with similar studies.

-Furthermore, the discussion should also highlight the benefit and limitations of the applied modeling techniques particularly when it comes to the implications of the current techniques in establishing priority zones for management.

Author Response

Response:

Thank you for reviewing our manuscript and for your valuable suggestions in pointing out its shortcomings. In response to your questions and suggestions, I have made the following recovery.

Introduction: Introduction does not provide sufficient review of the available geospatial modeling techniques, for example, available SDMs, particularly for the target species in the context of this study. In addition, previous studies on the target species on the context of this study has not been reviewed.

Objectives of the study should be clearly spelled out please.

Answers:

We add context to the species distribution model, summarize the main areas of study for the target species and a new clarification of the purpose of the study is provided. We have revised the text for relevant details and have retained the revision marks.

2 Materials and Methods

2.1. Data collection and processing

(1) The full link of the GBIF should be provided from where the records were downloaded.

(2) It’s not clear whether any spatial filtering or 'thinning' was performed for the presence records? In other word, what was the minimum distance between to occurrence points?

(3) Why Phase 5 (CMIP 5) was used in this study?  According to the “worldclim” website, the Representative Concentration Pathways (RCPs) are now obsolete and they recommend the use of the new version (CMIP6 data; i.e., the Shared Socio-economic Pathways (SSPs) in similar studies.

Answers:

(1) The exact URL of the download changes in real time and we are unable to show the correct URL, but we have shown the doi of the download in the references.

(2) The minimum distance between the points of occurrence is 1km.

(3) When downloading data in this study, CMIP5 data was commonly used in maxent model. Maybe I don't know enough. Combined with the species records of that time, we selected the CMIP5 data.

2.2. Species distribution modeling, optimization and evaluation

(1) "5 regularization multiplier value" this should be expressed as “(β)”

(2) “by default, the maximum number of iterations is set as 10000, “since the model was optimized, was more than or less than 10000 background points were tested?

(3) "AUC" Sometimes AUC alone is not sufficient to evaluate the model performance. Why TSS was not considered alongside the AUC?

Answers:

(1) We have added 'β' in brackets and changed 'RM' to 'β' in 'Table 2'.

(2) After the model has been optimized, the number of background points tested is more than 10,000.

(3) We have supplemented the evaluation system and numerical data of TSS in the materials and methods, results and discussion. Specific values can be found in Table 3 of the manuscript. Specific values can be seen in Table 3 of the manuscript. TSS of each period is greater than 0.80, indicating excellent performance of the model.

3 Results:

(1) Please refine/recreate the Figures 2, 3, and 5, hard to read.

(2) All the Figures need refinement please.

(3) I also suggest moving Figure 2 to the appendix or supplementary document.

Answers:

(1-2) Thank you for your important suggestions regarding our shortcomings. We have improved the high resolution of the images in the manuscript.

(3) Thank you for your valuable comments.

4 Discussion

The discussion section here in this paragraph is superficial and requires more work. Particularly when it comes to the in depth explanation of the key factors influencing the distribution of the target species. In other words, how environmental variables influence the spatial distribution of the target species is required. In addition, the discussion section requires drawing parallels with similar studies.

Furthermore, the discussion should also highlight the benefit and limitations of the applied modeling techniques particularly when it comes to the implications of the current techniques in establishing priority zones for management.

Answers:

(1) Thank you for your suggestions on the problems with the discussion section. We have made the appropriate changes including comparisons with similar studies and how the environment affects their distribution. All changes are reflected in the manuscript.

(2) Thank you for your valuable comments. We have discussed the strengths and weaknesses of the current model by comparing it with other research articles, and have marked the manuscript for revision marks.

Reviewer 2 Report

The paper “Prediction of potential suitable distribution areas of Quasipaa spinosa in China based on MaxEnt optimization model” is a study that models potential distribution of the ecologically and commercially important species, in the light of climate change. It points out the vulnerability of amphibians and importance of their conservation, with citing proper literature sources. The dataset is adequate for this type of study and statistical models are appropriate and up-to –date. The results also provide the valuable insight into distribution trends of this vulnerable species under different emission scenarios in the future. It is an important contribution to the field. Still, there are some presentation-related issues with the manuscript. The quality of figures is lacking, and the authors should provide figures in better resolution and with bolder lines, since many features of the figures are almost unreadable. The paper could also use some proofreading and I’ve noticed many grammar and phrasing issues in the written English, so I would wholeheartedly recommend the authors to send the paper to the professional English proofreading before the resubmission. Other than that, I would recommend this article to be accepted in Biology after a minor revision. Some more specific comments are provided in the annotated .pdf. file.

Author Response

Response:

Comments and Suggestions for Authors:

The paper “Prediction of potential suitable distribution areas of Quasipaa spinosa in China based on MaxEnt optimization model” is a study that models potential distribution of the ecologically and commercially important species, in the light of climate change. It points out the vulnerability of amphibians and importance of their conservation, with citing proper literature sources. The dataset is adequate for this type of study and statistical models are appropriate and up-to –date. The results also provide the valuable insight into distribution trends of this vulnerable species under different emission scenarios in the future. It is an important contribution to the field. Still, there are some presentation-related issues with the manuscript. The quality of figures is lacking, and the authors should provide figures in better resolution and with bolder lines, since many features of the figures are almost unreadable. The paper could also use some proofreading and I’ve noticed many grammar and phrasing issues in the written English, so I would wholeheartedly recommend the authors to send the paper to the professional English proofreading before the resubmission. Other than that, I would recommend this article to be accepted in Biology after a minor revision. Some more specific comments are provided in the annotated .pdf. file.

Answer:

Thank you for taking time out of your busy schedule to review our manuscript. We are very grateful for your comments and suggestions. We fully agree with the shortcomings in the manuscript that you have identified and are very grateful for your acknowledgement of our research. We have made careful corrections in the manuscript based on your suggestions and have marked the corrections.

Round 2

Reviewer 1 Report

The manuscript is sufficiently improved.